# Crystal-confined freestanding ionic liquids for reconfigurable and repairable electronics

Naiwei Gao[1], Yonglin He[1], Xinglei Tao[1], Xiao-Qi Xu[1], Xun Wu[1] & Yapei Wang[1]

Liquid sensors composed of ionic liquids are rising as alternatives to solid semiconductors for flexible and self-healing electronics. However, the fluidic nature may give rise to leakage problems in cases of accidental damages. Here, we proposed a liquid sensor based on a binary ionic liquid system, in which a flowing ionic liquid [OMIm]$PF_6$ is confined by another azobenzene-containing ionic liquid crystalline [OMIm]AzoO. Those crystal components provide sufficient pinning capillary force to immobilize fluidic components, leading to a freestanding liquid-like product without the possibility of leakage. In addition to owning ultra-high temperature sensitivity, crystal-confined ionic liquids also combine the performances of both liquid and solid so that it can be stretched, bent, self-healed, and remolded. With respect to the reconfigurable property, this particular class of ionic liquids is exploited as dynamic circuits which can be spatially reorganized or automatically repaired.

[1] Department of Chemistry, Renmin University of China, Beijing 100872, China. Correspondence and requests for materials should be addressed to Y.W. (email: yapeiwang@ruc.edu.cn)

Terminator 2: Judgment Day was one of the most popular science fiction movies in the last century. The movie told about an intelligent liquid robot of T-1000 that could be reshaped to pretend as different people and rapidly repaired once it was mechanically damaged[1–3]. This terminator robot has stayed in the imagination for 27 years since the movie was released in 1991. In principle, such a liquid robot should be served by special liquid electrical components with intrinsic advantages of high deformability[4], self-healing[5–8] and reconfigurable[9,10] abilities. However, it is still the main ideology to formulate electronic devices only by solid electrical components with fixed metallic tracks and semiconductor units covered on silicon or polymer substrates[11–16]. Interests were raised to exploit liquid conductors, such as liquid metallic alloys with negligible electrical resistance, as fluidic circuits for flexible or self-healing electronic devices[17–20]. Beyond the conductive purpose, great efforts were also devoted to exploiting liquid sensing materials, which may enable the generation of liquid sensors for more complicated circuits[21–24]. A breakthrough was recently achieved by the innovation of thermal sensing ability of ionic liquids[25,26]. This particular class of liquid "semiconductors" was further extended as light detectors and pressure sensors[27]. Nevertheless, liquid electronic materials up to present are still far from practical use for fabricating reliable circuits, not to mention liquid robots. Liquid materials have to be encapsulated in vessels because they are lack of free-standing properties. The solid encapsulating layers hardly obey the fluidic nature of inner liquids. Leakage problems may be also encountered in the case of accidental damages.

A few systems have been exploited to confine fluidic electronic materials within a solid matrix by capillary effect, and hence prevent liquid leakage when the encapsulating layers are broken. In order to attain stronger capillary force thus enhance the interfacial adsorption, it is necessary to reduce the interstitial size of the encapsulating matrix according to the capillary theory. A general method is to load guest liquids in host micro-channels by which the loading capacity at open states is inversely proportional to channel diameter[25]. Similar to micro-channels, porous materials could also serve as host matrix to accommodate liquid materials and the capillary force was readily improved by minimizing pore size[27]. However, practical examples of host materials fitting the high degree of deformation and physical rupture of loaded liquids are sparse due to the lack of satisfactory elasticity and self-healing ability. Freezing the liquid at a temperature below melting point is considered as another strategy to solve leakage problem[28], yet this strategy rarely raises attention in electronics because the fluidic advantages are completely lost. The phase transition from a liquid state to a solid state, on the other hand, may get rid of the encapsulating layer, which removes the sensing barrier and reinforces the electrical transport. In these regards, the T-1000 robot is more like a complex of liquid materials and phase-transition materials, which is expected to possess the free-standing[29] character of solid materials and flowing performance of liquid materials. As such, it can be arbitrarily deformed or rapidly repaired to counter the mechanical damage and leakage problem.

Aiming to exploit liquid electronics like Terminator robot, we proposed an ionic liquid binary system—crystal-confined ionic liquids (CCILs) in which an ionic liquid is confined in a crystallized ionic liquid. This system possesses the intrinsic fluidic properties of high flexibility, self-healing, and reconfigurable abilities. At the same time, it also has a solid-like function of free-standing character without external encapsulating protection and mechanical support.

## Results

### Preparation and characterization of CCILs. To be specific, 1-methyl-3-octyl-1H-imidazolium (E)-4-(phenyldiazenyl)

phenolate ([OMIm]AzoO) as an ionic liquid[30] containing azo-benzene group was synthesized through the classical strategy (Supplementary Fig. 1 and Supplementary Fig. 2). This ionic liquid is easy to crystallize into rectangular fibers owing to 1-octyl-3-methylimidazolium hexafluoro-phosphate ([OMIm]PF$_6$) was chosen as another type of ionic liquid which has the same cation with [OMIm]AzoO. Less than 3 wt.% of [OMIm]AzoO could be dissolved in [OMIm]PF$_6$ at room temperature (Supplementary Fig. 3), while two ionic liquids are fully miscible with each other at the temperature above the melting point of [OMIm]AzoO. After heating-cooling for two cycles as illustrated in Fig. 1a, two kinds of ionic liquids formed into a non-flowing product (Fig. 1b). The dark brown color of the mixed ionic liquids is attributed to the characteristic absorption of azobenzene moiety of [OMIm]AzoO in the visible light window. As shown in Fig. 1c, the melting point of [OMIm]AzoO is 70 °C according to DSC analysis and it is downshifted upon addition of [OMIm]PF$_6$. The only one phase transition temperature reveals that [OMIm]AzoO and [OMIm]PF$_6$ are fully miscible at melting state. It is believed that the same cation excipient accounts for the outstanding compatibility between two kinds of ionic liquid. Therefore, changing the ratio between [OMIm]AzoO and [OMIm]PF$_6$ can enable the regulation of phase transition temperature, which allows the CCILs binary system to be switched between the solid state and liquid state at the desired temperature (Supplementary Fig. 4). It is worth noting that negligible decomposition is observed for CCILs at a temperature below 200 °C (Supplementary Fig. 5). The excellent thermal stability ensures the sensing applications in a wide temperature range.

Though two kinds of ionic liquid have outstanding compatibility, [OMIm]AzoO is crystallized and separated out from [OMIm]PF$_6$ once the temperature drops below the melting point. To view the molecular packing at solid state, the [OMIm]AzoO crystal grown from a mixed solvent of hexane and chloroform was characterized by single crystal X-ray diffraction (Fig. 1d). Surprisingly, the distance between the benzyl moieties ring is 6.48 Å which exceeds the distance of π-π interaction. The distance between the benzyl ring and imidazole ring is 3.87 Å, which is located within the distance accounting for effective π-π stacking[31] (Supplementary Fig. 6a and Supplementary Dataset). Namely, the driving force for the crystallization should be due to the intermolecular π–π interaction[32,33] between azobenzene group and imidazole cation. In order to confirm the crystal structure of [OMIm]AzoO that is grown in [OMIm]PF$_6$, the crystals as a form of rectangular fibers were characterized by regular powder X-ray diffraction (XRD). The [OMIm]AzoO has poor solubility in [OMIm]PF$_6$, as stated above, which is nearly separated out as a form of crystal fibers at a temperature below the melting point. As a consequence, a greater amount of crystals were formed upon loading more [OMIm]AzoO in [OMIm]PF$_6$. Characteristic peaks assigned to crystallized [OMIm]AzoO exist in all crystal samples, indicating [OMIm]PF$_6$ has no effect on the molecular packing of [OMIm]AzoO. Additionally, theoretical XRD calculation based on the single crystal X-ray diffraction result fully agrees with the experimental powder XRD patterns (Supplementary Fig. 6b-d), which further confirms the solvent independent crystallization of [OMIm]AzoO.

### Theoretical calculation of maximum adsorption of CCILs.
Though [OMIm]PF$_6$ does not take part in the crystallization of [OMIm]AzoO, it is assumed that [OMIm]PF$_6$ is able to wet the [OMIm]AzoO crystals in terms of the same cation species. In this case, the [OMIm]AzoO crystal fibers should provide pinning capillary force to anchor fluidic [OMIm]PF$_6$[34,35]. Following the optical microscope observation (Fig. 2a), the crystal fibers are

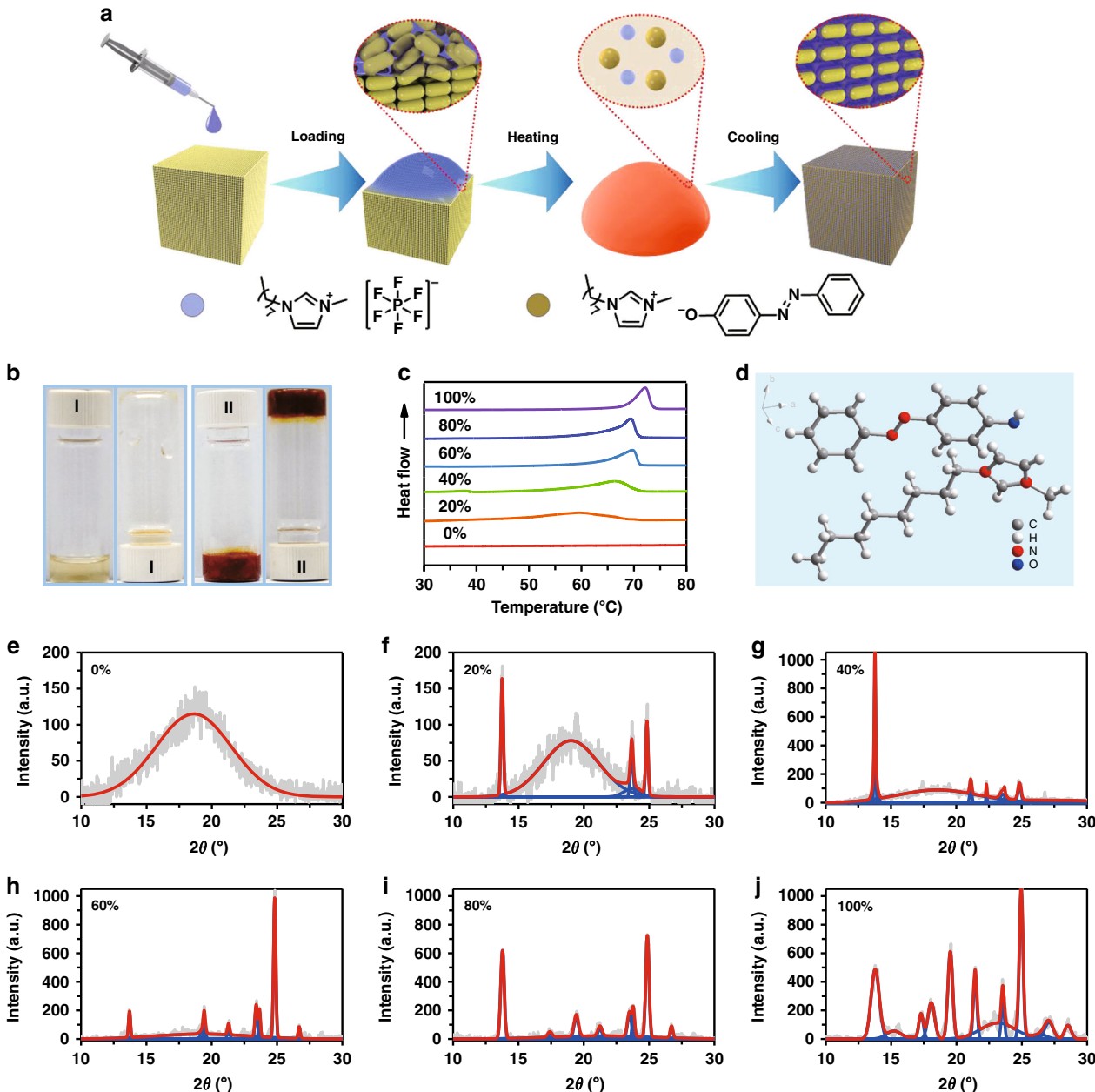

**Fig. 1** Scheme of preparation and characterization of CCILs (crystal-confined ionic liquids). **a** Preparation of CCILs as a complex of [OMIm]PF$_6$ and [OMIm]AzoO through a super-saturated solution cooling method. **b** Photos of [OMIm]PF$_6$ (I) and CCILs with addition of 60 wt.% of [OMIm]AzoO (II). **c** DSC curves of [OMIm]PF$_6$ and CCILs loading different amounts of [OMIm]AzoO (20, 40, 60, 80, 100 wt.%). **d** Crystal structure of [OMIm]AzoO obtained by single crystal X-ray diffraction. **e-j** Powder XRD analysis of CCILs loading different amounts of [OMIm]AzoO. **e** 0 wt%, **f** 20 wt%, **g** 40 wt%, **h** 60 wt%, **i** 80wt.%, **j** 100 wt%) (gray line: raw data; red line: enveloping lines; blue line: imitating peaks)

gradually condensed as the increasing amount of [OMIm]AzoO in [OMIm]PF$_6$. Individual fibers are hardly observed when the weight concentration of [OMIm]AzoO is over 40%, as the crystal fibers are arranged in a compact manner. In practice, [OMIm]AzoO has four states in [OMIm]PF$_6$ solution, including soluble state, suspended state, loose are possible to cause the rheological change of [OMIm]PF$_6$ based on capillary effect. As shown in Fig. 2d, the CCILs with different amounts of [OMIm]AzoO were encapsulated in glass tubes with a diameter of 3 mm and height of 5 cm. Pure [OMIm]PF$_6$ rapidly flows out of the glass tube owing to the gravity effect. The CCILs with the addition of 20 wt.% [OMIm]AzoO or more, nevertheless, are spatially confined in the glass tube without any leakage for a long term. This stop-flowing phenomenon convinces that the pinning capillary force supplied

by [OMIm]AzoO crystal fibers, rather than the capillary force provided by the glass tube, is critical to confine fluidic [OMIm]PF$_6$ in an open space. The apparent capillary force is the sum of capillary force that is provided by each crystal fiber. In this regard, the apparent capillary force may reach maximum only when each [OMIm]AzoO crystal fiber is fully wetted by [OMIm]PF$_6$. In terms of loose accumulation, the morphology of crystal fibers at 20 wt.% concentration was statistically analyzed by confocal microscopy observation (Supplementary Fig. 7), which is approximatedas rectangular shape regardless of the length. However, it is challenging to view the real packing feature of [OMIm]AzoO crystal fibers in [OMIm]PF$_6$ phase. According to the microscopic observation, crystal fibers are nearly aligned in one direction in the case of close packing. The interstitial spaces

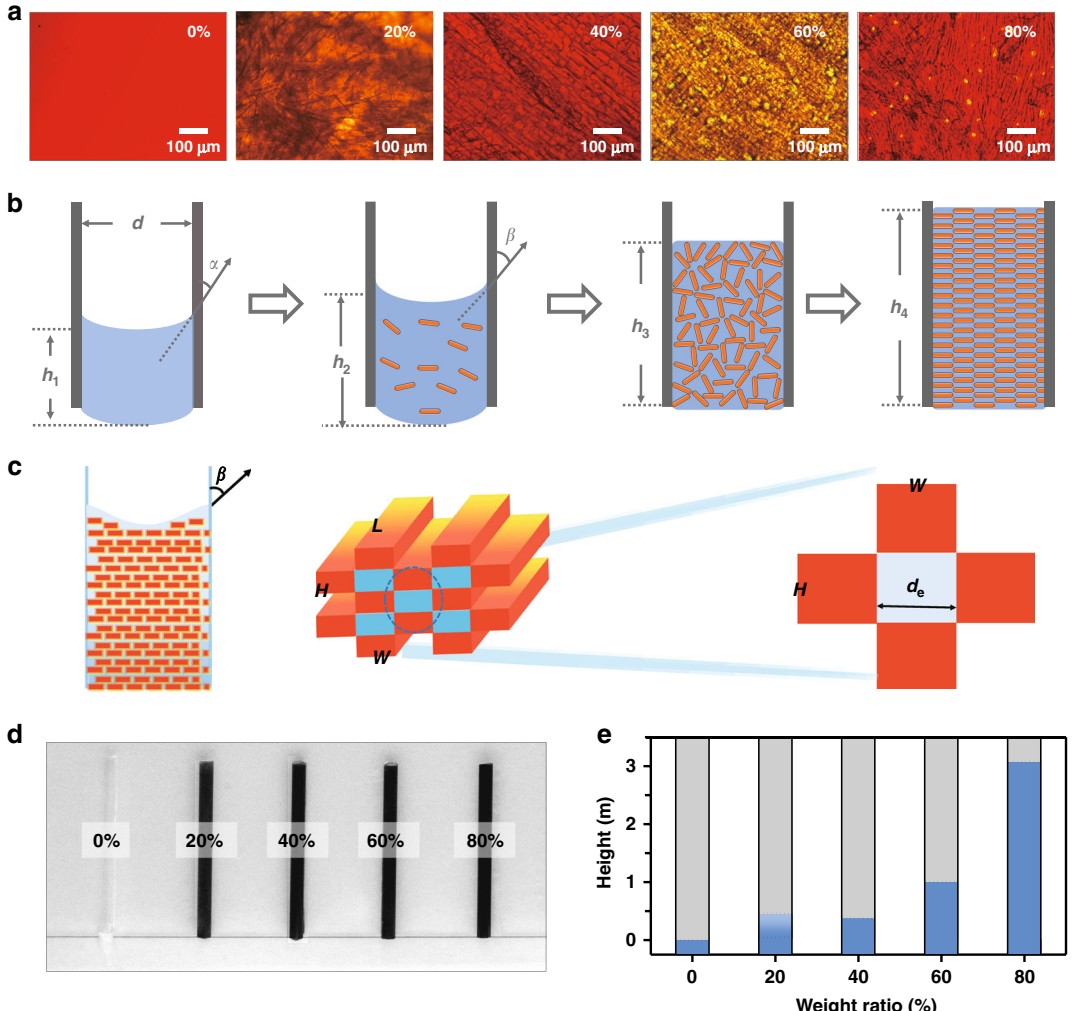

**Fig. 2** Prediction and measurement of CCILs leakage from glass tubes with open ends. **a** Microscopic images of CCILs containing different contents of [OMIm]AzoO. Scale bar: 100 μm. **b** Schematic illustration of the different status of [OMIm]AzoO in [OMIm]PF$_6$, including solution state, suspended state, loose accumulation state and close-packed condition (from left to right). *h*: height of ILs in the glass tube; *d*: diameter of the glass tube; *α*: contact angle between the ILs and glass tube; *β*: contact angle between CCILs and glass tube. **c** A mathematical model demonstrating the pinning capillary effect by [OMIm]AzoO crystals. **d** Experimental demonstration of retention of CCILs with different contents of [OMIm]AzoO upon placing the glass tubes vertically (glass tubes have diameter of 3.0 mm and length of 50.0 mm). **e** The theoretical values of the maximum height of the CCILs with different contents of [OMIm]AzoO that the glass tube can hold without leakage phenomenon

among the crystal fibers are filled with [OMIm]PF$_6$. Intrinsically, the confinement of the mixture of [OMIm]PF$_6$ and [OMIm]AzoO crystal fibers is due to the existence of Laplace pressure across the liquid-air interface. Here, in order to calculate the theoretical adsorption of CCILs, we proposed to replace the inner capillary with an equivalent diameter. As shown in Fig. 2b, c, Supplementary Equation 1–12, and Supplementary Fig. 9, we can calculate the equivalent diameter according to Eq. 1,

$$d_e = \frac{4\varphi_L}{\rho_s \varphi_S \sigma} \tag{1}$$

Therefore, the theoretical maximum adsorption can be achieved when the crystals do not touch with each other, as shown in Eq. 2,

$$\sigma \leq \frac{2(WH + WL + HL)}{WHL \times \rho_s}. \tag{2}$$

The theoretical height of [OMIm]PF$_6$ loading 20 wt.% [OMIm]AzoO or more that can be confined in a glass tube is predicted by Eq. 3,

$$h_{max} = \frac{2\gamma_{gl}\cos\theta}{g} \times \frac{m_s}{m_l} \times \frac{WL + WH + HL}{WHL\rho_s}, \tag{3}$$

where $\gamma_{gl}$ is the surface tension of saturated solution of [OMIm]AzoO in [OMIm]PF$_6$. $\theta$ is contact angle between crystals and the saturated solution of [OMIm]AzoO in [OMIm]PF$_6$. $m_l$ and $m_s$ are the mass of [OMIm]PF$_6$ and [OMIm]AzoO. $\rho_s$ is the density of [OMIm]PF$_6$. $L$, $W$, and $H$ refer to the length, width, and height of the crystals, respectively (see all the details in Supplementary Fig. 8 and Supplementary Note 1).

The theoretical confinement height of [OMIm]PF$_6$ loading 20 wt.% [OMIm]AzoO is estimated as 0.3048 m. It is difficult to predict the confinement height of [OMIm]PF$_6$ loading extremely close-packed [OMIm]AzoO crystal fibers because the crystal size cannot be accurately measured in situ. However, it is convinced that the size of the crystal fibers, including length ($L$), width ($W$) and height ($H$), should decrease owing to the increased number of crystal nucleus as well as the decreased space for crystal growth. The size decrease is beneficial for improving the value of

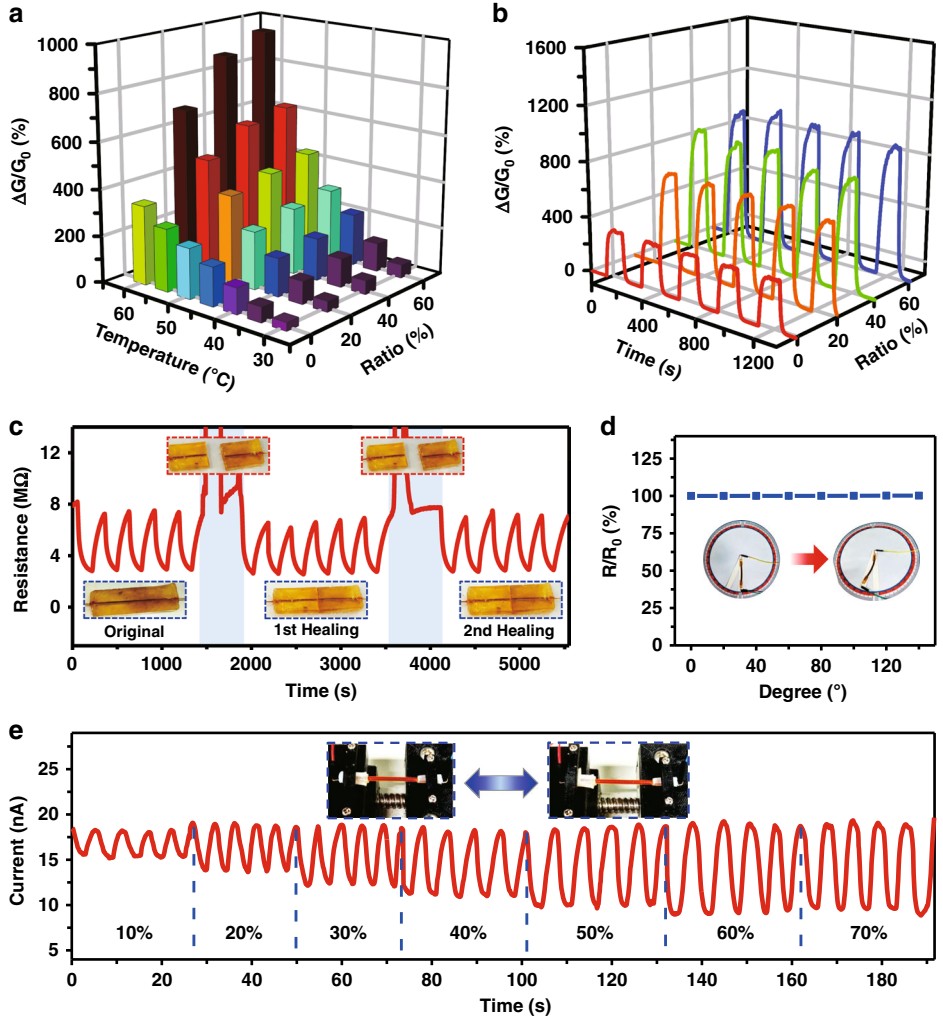

**Fig. 3** Electrical sensing performance of CCILs. **a** Relative conductivity change of CCILs with different contents of [OMIm]AzoO at different temperature ($\Delta G/G_0$ is the relative conductivity change, $\Delta G = G-G_0$). **b** On–off cycles of thermal response of CCILs operated between 60 °C and room temperature (25 °C) (red, orange, green and blue lines represent CCILs loaded with [OMIm]AzoO of 0, 20, 40, 60 wt%, respectively). **c** The process of breaking and healing of CCILs with an addition of 60 wt.% [OMIm]AzoO. Temperature is changed from 25–40 °C. The CCILs are sealed in a microchannel made by self-healing polymer. **d** Resistance change of CCILs filled in an elastic capillary tube at different bending angles. **e** Strain response of CCILs filled in a stretchable silicone tube

confinement height according to Equation 3. On the other hand, the increase of $m_s$ and decrease of $m_l$ upon adding more [OMIm]AzoO in [OMIm]PF$_6$ also tailors the improvement of confinement height. As summarized in Fig. 2e, without counting the size change in the calculation, only the increase of [OMIm]AzoO weight dramatically improves the theoretical height. Setting the size as the same as 20 wt.% [OMIm]AzoO crystal fiber, the confinement heights are 0.8122 m, 1.8275 m and 4.8732 m for [OMIm]AzoO with weight concentration of 40 wt.%, 60 wt.% and 80 wt.%, respectively. It should be noted that the theoretical adsorption height is irrelevant to the diameter of the glass tube, suggesting that crystal-confined ionic liquids are free-standing products and may be used without the external encapsulation.

**Sensing and self-healing performance.** Previous studies have revealed some particular ionic liquids can behave like a fluidic "semiconductor"[36,37]. Their conductivity increases upon increasing the environmental temperature, which is attributed to the acceleration of ionic mobility at a higher temperature. Though the crystal confinement raises the possibility to serve fluidic [OMIm]PF$_6$ as free-standing sensing materials, it, on the other

hand, impairs the ionic mobility so that the conductivity of [OMIm]PF$_6$ will be influenced (Supplementary Fig. 10). The electrical current is not evident for the binary ionic liquid system consisting of 80 wt.% [OMIm]AzoO. In this case, it is believed that too many crystal fibers will decrease the number of pathways and thus weaken the ionic migration. The impedance-time (IMPT) mode was chosen for conductance and sensing tests to reduce the existence of electrode polarization of ionic double layers that are generally appeared by direct current measurement (Supplementary Fig. 11). Upon adding [OMIm]AzoO with proper ratio to ensure the conductivity, the binary ionic systems possess the sensing ability against the temperature change. The thermal sensitivity of [OMIm]PF$_6$ loaded with different amounts of [OMIm]AzoO is compared in Fig. 3a. The sensitivity is positively relevant to the weight ratio of [OMIm]AzoO. Specifically, the conductivity change reaches 890% for the ionic liquid with an addition of 60 wt.% of [OMIm]AzoO, when the temperature rises from room temperature (25 °C) to 60 °C. This value is the largest thermal response as a form of conductivity change for organic sensing materials so far as we know. As explained before, the thermal response of pure [OMIm]PF$_6$, which follows

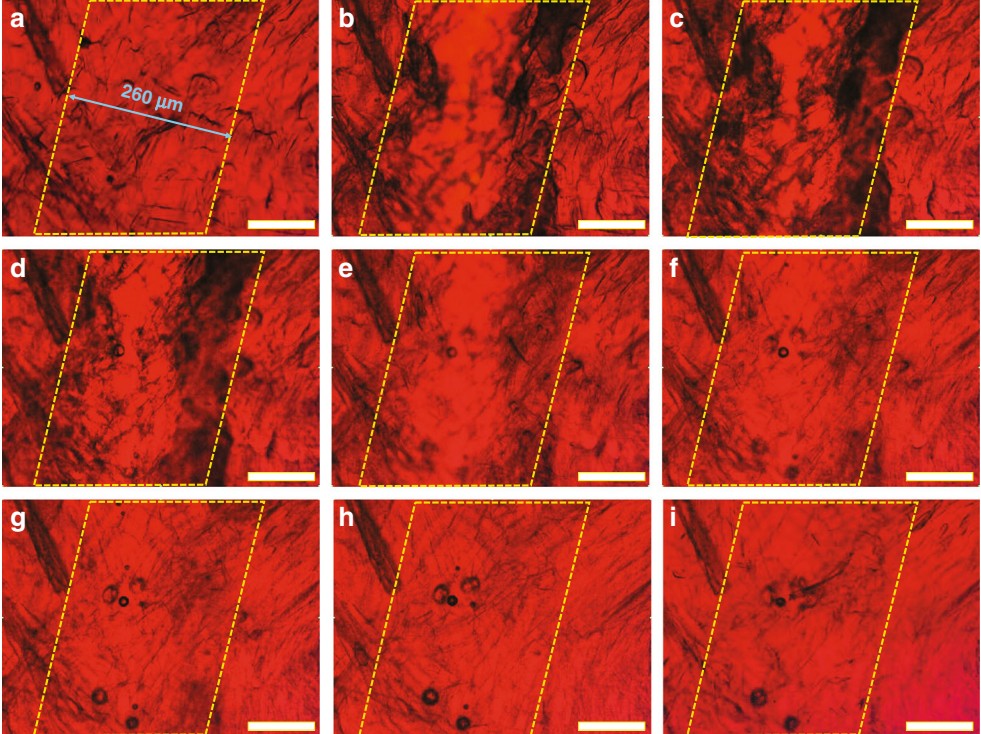

**Fig. 4** Microscope observation of the self-healing process at a micron-size scale. **a** Original CCILs loaded with 60 wt.% [OMIm]AzoO (25 °C). **b** Damaged sample (25 °C). **c–g** Raising temperature from 25 to 60 °C. **h**, **i** Cooling temperature to 25 °C. Scale bar: 100 μm

Vogel-Tamman-Fucher equation[38,39], is attributed to the increase of ionic mobility as a result of the reduced viscosity upon raising the temperature. The addition of [OMIm]AzoO crystal fibers can tailor a greater conductivity change because the number of conductive pathways and free ions is dramatically increased once the crystal fibers melt at high temperature. As shown in Fig. 3b, reproducible on–off electrical responses between room temperature and high temperature (60 °C) ensure the satisfactory reliability of these crystal-confined ionic liquid sensors. The outstanding repeatability was further confirmed by 100 cycles of the thermo-sensing tests of CCILs with different content of [OMIm]AzoO (Supplementary Fig. 12). Such an intriguing reliability under alternating heating and cooling treatment, namely, suggests that the repeated melting and crystallization of [OMIm]AzoO have a negligible impact on the long-term sensing performance.

Although [OMIm]PF$_6$ is confined within [OMIm]AzoO and their mixture is like solid, attractive advantages of fluidic materials including self-healing and highly flexible properties are well retained. Taking the CCILs containing 60 wt% [OMIm]AzoO as an example, an electrical sensor by sealing ionic liquids in a self-healing polymer channel exhibits excellent self-healing performance. As shown in Fig. 3c, the sensor breaks down when it is cut into two separate parts. However, the sensing ability against temperature changes immediately recovers once two separate parts touch each other. Such a fast sensing recovery demonstrates that [OMIm]PF$_6$ confined by crystal fibers should have preserved its fluidic nature. The fluidic nature was further convinced by the highly flexible and stretchable performance of CCILs. The conductivity keeps almost the same for the CCILs encapsulated in an elastic tube which was bent at different angles (Fig. 3d). Tensile stretching results in the decrease of conductivity (Fig. 3e) because the resistance increased at stretching states according to the ohm law. The reversible conductivity change under alternating tensile stretching and relaxation renders the

possibility to extend CCILs as strain sensors. Physical damage of crystal fibers that may happen during mechanical bending and tensile stretching does not have an effect on the conductivity of CCILs. A model has been supplied to explain the independence of electrical resistance on the length of crystal fibers at a given [OMIm]AzoO weight ratio. It is noteworthy that the electrical performance of CCILs against mechanical manipulation and self-healing treatment is well preserved at a smaller scale. According to electrical tests in Supplementary Fig. 13, the CCILs loaded within a micro-channel endure mechanical bending at an arbitrary bending angle, meanwhile, keeping the electrical resistance unchanged. The electrical conductivity, as well as thermo-sensing ability, is almost fully recovered after many cycles of cutting and healing operations.

In order to discern the self-healing mechanism of CCILs, a typical self-healing process of CCILs within a specific area was recorded under microscope observation. As shown in Fig. 4a, b, at the moment of suffering injury, the ionic pathway was cut off due to the disconnection of [OMIm]AzoO crystals, which initially serve as capillary fillers to confine and bridge [OMIm]PF$_6$. Notably, the self-healing process began with the migration of ionic liquid of [OMIm]PF$_6$ into the wounded area upon increasing temperature (Fig. 4c, d). This self-motivated flowing phenomenon is attributed to the weakened confinement of [OMIm]AzoO crystal fibers when they are partially melted at an increased temperature. In the next stage, more and more ionic liquids, driven by the melting process of [OMIm]AzoO crystal fibers, were directed to the wounded area (Fig. 4e–g), immediately followed by the migration of some unmelted [OMIm]AzoO crystals together with [OMIm]PF$_6$ into the wounded area. These migrated crystal fibers acted as "seed" crystals, leading to the growth and regeneration of [OMIm]AzoO crystal fibers in the wounded area. The crystallization of [OMIm]AzoO was accelerated when the temperature was dropped to room temperature (Fig. 4h, i) and the increased number of [OMIm]AzoO crystal

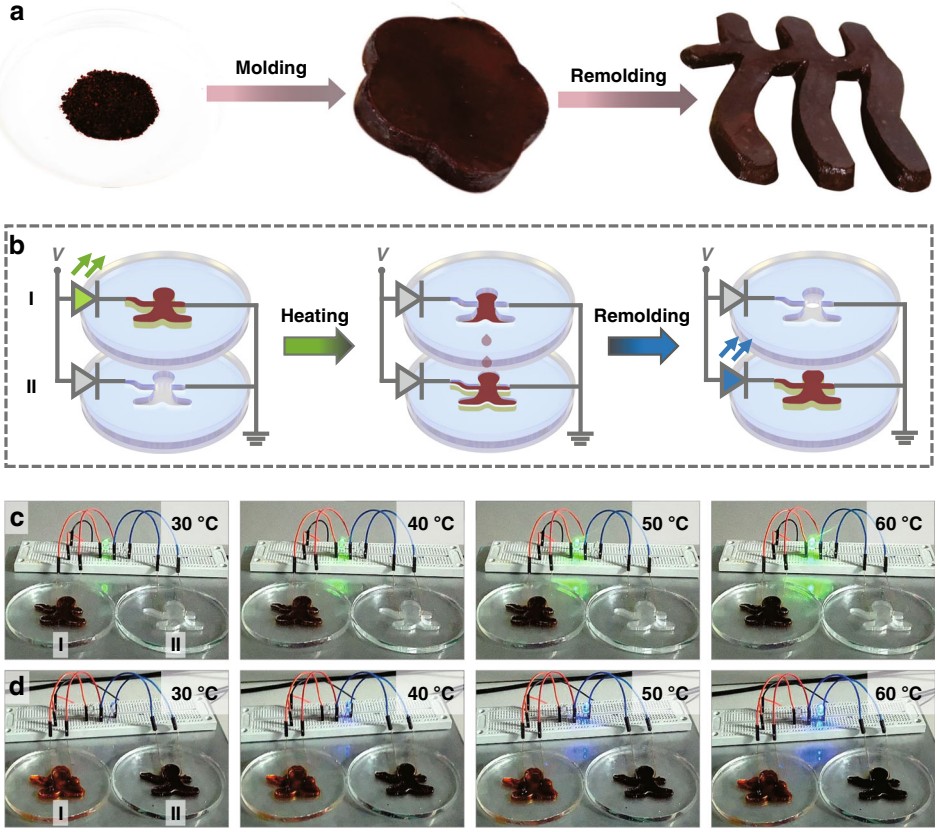

**Fig. 5** Freestanding and reconfigurable measurements. **a** Freestanding CCILs (60 wt.% of [OMIm]AzoO) with different shapes, from left to right: powder-like product, plum blossom, and simplified logo of the Renmin University of China. **b** Scheme of the transport of CCILs containing 60 wt.% of [OMIm]AzoO from the upper circuit to the bottom circuit. **c** Thermal response of the upper circuit indicated by a green LED at different temperatures (30, 40, 50, 60 °C). **d** The thermal response of the bottom circuits indicated by a blue LED at different temperatures (30, 40, 50, 60 °C)

fibers recovered to confine [OMIm]PF$_6$. At this stage, [OMIm] PF$_6$ stopped flowing and the broken ionic pathway was fully healed. Based on the in-situ microscopy observation of the self-healing process of CCILs, it is considered that the self-healing process consists of two main steps. The first step is the migration of [OMIm]PF$_6$ together with some [OMIm]AzoO crystal fibers from the bulk phase to the wounded area. The second step is the recrystallization of [OMIm]AzoO crystals which confine [OMIm] PF$_6$ and repair the wounded area. Besides, AFM study also gives a similar demonstration on the self-healing mechanism, though it indicates the self-healing process is faster at the smaller scale (Supplementary Fig. 14).

**Characterization of reconfigurability**. As stated above, the confinement height, in principle, is not relevant to the diameter of the encapsulating glass tube. In this regard, CCILs may be used as a particular class of free-standing materials yet possess natural fluidic properties. The [OMIm]AzoO crystals likely act as a framework to immobilize fluidic [OMIm]PF$_6$ based on capillary effect. Such a binary system is reconfigurable upon melting and recrystallizing [OMIm]AzoO crystals. As shown in Fig. 5a, the CCILs containing 60 wt.% [OMIm]AzoO could be molded into a free-standing sample with arbitrary shapes by increasing temperature above the melting point and then cooling to room temperature. It is evidently free-standing without any deformation when it is placed on the bench or grabbed in hand (Supplementary Fig. 15). The free-standing phenomenon means that the improved interfacial interaction based on crystal confinement is strong enough to overcome the gravity-induced flowing of ionic liquids. For example, a plum blossom shape was easily remolded

into the logo of our university. Thus, CCILs can be used as solid-like electronic circuits without worrying about leakage problem. In terms of the wonderful reconfigurable property, CCILs were exploited as a dynamic electrical sensor, which could be spatially transferred from one circuit to another circuit. As illustrated in Fig. 5b, two isolated circuits were built in which a green and a blue LED was installed separately. Each circuit contains a mold with human-like shape and the two molds were placed up and down in parallel. When the upper mold is filled with CCILs, the conductance within the corresponding circuit is improved upon increasing the temperature, which is represented by the enhanced brightness of the green LED (Fig. 5c). Once the temperature is raised above 70 °C at which [OMIm]AzoO crystals are melt and capillary force is disappeared, CCILs flow from the upper mold into the bottom mold due to gravity effect (Supplementary Movie 1). As a consequence, the upper circuit was disconnected, and meanwhile, the bottom circuit was connected. The indicator of blue LED was gradually brightened with the increase of temperature, indicating the success of the sensing function of CCILs in the bottom circuit (Fig. 5d). In addition to the application as dynamic electrical sensors, the reconfigurable property also offers great opportunities for recycling the sensing materials and thus reducing electrical wastes.

**Self-healing and reconfigurable artificial robot arm**. In order to demonstrate the potential of CCILs for robot design, an artificial arm made of CCILs containing 60 wt% [OMIm]AzoO was attached on a wooden doll (Fig. 6a). In this special arm, a controlling circuit was implanted to probe mechanical damage and trigger automatic repair (Supplementary Movie 2). As illustrated

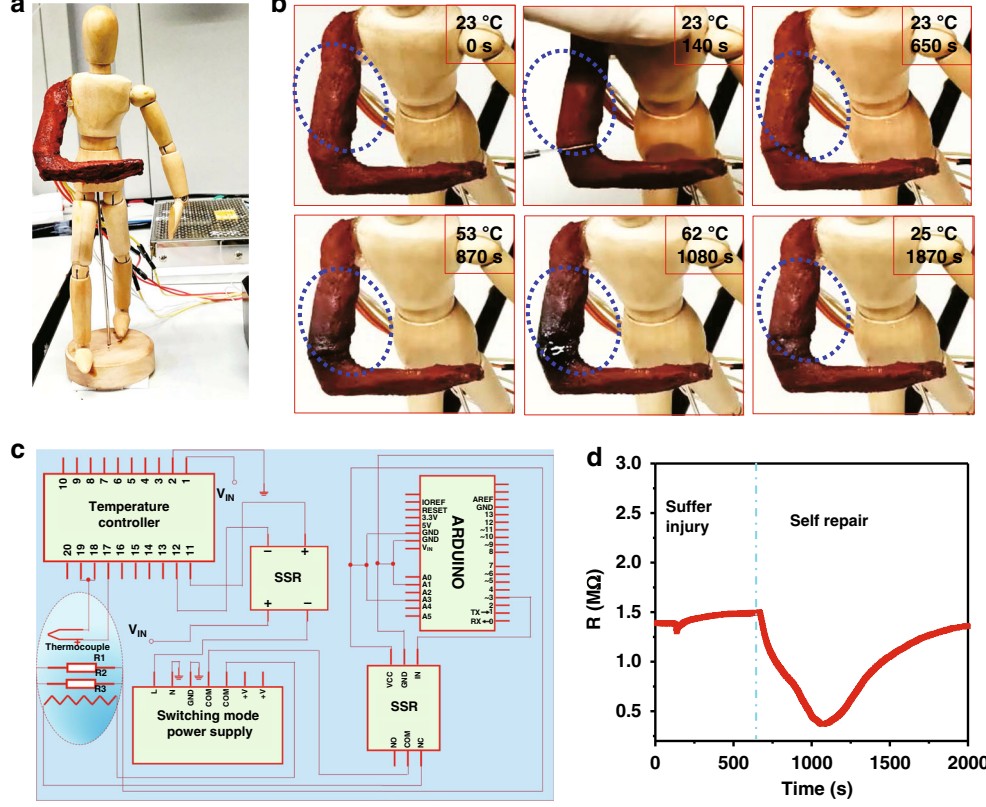

**Fig. 6** A self-healing robotic arm made of CCILs. **a** A wooden doll installed with a CCILs (60 wt.% of [OMIm]AzoO) arm connected with an intelligent temperature controlling system. **b** The process of cutting and healing of the robotic arm at different status: 0 s without injury; 140 s with injury; 140–1870 s self-healing behavior with manual intervention. **c** The diagram of a controlling circuit for probing and repairing mechanical damages. **d** Real-time resistance monitoring of the robotic arm at each state according to the self-healing process (**b**)

in Fig. 6c, this controlling circuit consists of three interconnected parts—a resistance monitoring part, a temperature controlling part, and a heating part. Specifically, the resistance monitoring part plays a role to monitor the resistance change of the robotic arm. Once the resistance of the robotic arm is increased above a threshold of 150 MΩ, it arouses the temperature controller followed with local heating by the thermal heater. The thermal heating stops whenever the resistance drops below the threshold. According to the phase transition study, the pure [OMIm]AzoO crystal has a melting point of about 70 °C. Hence the temperature controller is input an order with a maximum temperature as 70 °C, above which the circuit is cut off. As shown in Fig. 6b, d, the robotic arm successfully completed a self-healing process without manual intervention. Before receiving mechanical damage, the arm has a constant resistance. Once the robotic arm is heavily cut by a sharp blade, the resistance of robotic arm drops immediately. The decrease of resistance is due to the insertion of the metallic blade which has a much smaller resistance in comparison with CCILs. Then, the resistance of robotic arm monitored by a single chip microcomputer increases after the removal of the blade, which exceeds the threshold of 150 MΩ. The temperature controller was then turned on, leading to local heating at the damaged region. The increase in temperature is consistent with the decrease of resistance, which is attributed to the thermal-sensitive performance of CCILs as stated above. With further increase of temperature, phase transition happens and CCILs are reconfigured at the damaged region. The wound on the robotic arm caused by the blade cutting was fully healed in several minutes. When the resistance reduces to 40 MΩ, the temperature controller releases another order to terminate the thermal heating

process and the healed robotic arm is cooled down to room temperature.

**The choice of crystal-confined ionic liquids.** According to corresponding calculation, the fiber-like crystals of [OMIm]AzoO provided pinning capillary forces to anchor another ionic liquid of [OMIm]PF$_6$ and thus prevented liquid flowing even at the states without external encapsulation. The crystal-confined phenomenon, which is the key characteristic for directing the formation of free-standing liquids, is hardly observed in other binary-phase ionic liquid systems. As summarized in Supplementary Fig. 16, upon mixing with [OMIm]PF$_6$, four common solid ionic liquids ([BMIm]Cl, [VEIm]PF$_6$, [EMIm]PF$_6$, [VEIm]Br) with different ion sizes and ion species only generated either a solution-based mixture or phase-separated complex with precipitation. In practice, if two ionic liquids have good compatibility, one is fully dissolved in another. Otherwise one is precipitated and separated from another. To overcome the gravity-induced liquid flow, one ionic liquid is expected to evolve into particular fillers, which still keep excellent compatibility with another ionic liquid and provide an interfacial attraction to confine the liquid phase. Those fillers should be reconfigurable, like reversible melting and crystallization as demonstrated, offering promises for recycling the binary ionic liquid system. The binary ionic liquid system is unlike the direct addition of solid nano-fillers to prevent liquid flow, which can be considered as a strategy of heterogeneous complexation. It generally needs special treatments to improve the dispersity of nano-fillers and also has the problem of difficult reconfiguration[5,40–47]. In this work, the wonderful miscibility of two fluidic ionic liquids at high

temperature ensures the formation of well-dispersed crystal fibers of [OMIm]AzoO in [OMIm]PF$_6$. There should be no special interactions among [OMIm]AzoO crystal fibers and three-dimensional networks are not anticipated. However, in terms of having the same cations, [OMIm]AzoO crystals should provide substantial interaction with [OMIm]PF$_6$, namely, [OMIm]AzoO crystals are well wetted and bridged by [OMIm]PF$_6$. Whenever the concentration of [OMIm]AzoO is high enough, at which the crystal fibers are closely packed, the capillary effect is significantly improved to block the flow of [OMIm]PF$_6$ as a result of narrowing the interstitial space between crystal fibers. Intrinsically, the ionic liquid complex is free-standing, and at the same time, liquid-like based on lateral sliding of crystalized ionic liquid together with the free ionic liquid. As demonstrated in Supplementary Fig. 17, the case of larger loss modulus than storage modulus by rheology tests is indicative of the fluidic nature of CCILs.

Supramolecular polyelectrolyte hydrogels could also be potentially used as conductive materials for self-healing and recyclable electronics. There have been already many excellent examples of exploiting hydrogels as flexible and self-healable electronic sensors[48–50]. However, it is worth noting that hydrogel systems are routinely encountered with the problems of water volatilization at open states, water swelling in high humidity conditions, or frozen at a low temperature. These disadvantages definitely hinder the long-term service of the hydrogel systems in the complex environment. Ionic gel solely composed of ionic liquids and gelators is a better choice to avoid the problem of solvent volatilization. In the past decade, intensive studies were devoted to self-assembly of block copolymers for confining ionic liquids, especially thermo-reversible self-assembly of block copolymers that can trigger the sol-gel transition[51,52]. To our knowledge, the study of ionic liquid-containing gels is still at a young stage. Rarely ionic liquid gel systems were extended to sensing applications. In our opinions, they may be compatible with reconfigurable electronics if the reassembly of polymer gelators has little influence on the conductive and sensing performance, meanwhile, keeping the content of gelator at a reasonable value.

## Discussion

In summary, we presented a thermal-sensitive material that was formulated by confining one ionic liquid within another ionic liquid crystal based on substantial capillary effect. This binary ionic liquid system acts like a free-standing fluid, possessing fluidic properties yet without leakage problem. The theoretical analysis of capillary force gave an explanation that the crystal-confined effect is a result of the tremendous increase of liquid/solid interface. In terms of fluidic performance, this particular sensing material has shown great potentials in the preparation of highly flexible and self-healing electronic sensors. Reversible melting and crystallization afford great convenience to recycle and reconfigure such a sensing material. These unique advantages lay the groundwork for developing green electronics with less production of electronic wastes. Further successful exploitations as dynamic circuits and self-repairing arm demonstrate that the concept of free-standing liquid materials opens an avenue to generate smarter electronic products, like T-1000 robot in the movie of Terminator. It is envisioned that ionic liquids may be also confined in other stimuli-responsive crystals to trigger the phase transition in different ways.

## Methods

**Materials**. Phenol (AR), Aniline (ACS, 99.0%), potassium hydroxide (AR), urea and Ammonium chloride (PT) were purchased from Aladdin Industrial Corporation. 1-octyl-3-methylimidazolium chloride ([OMIm]Cl, 99%) and 1-octyl-3-methylimidazolium hexa-fluoro-phosphate ([OMIm]PF$_6$, 99%) were purchased

from Lanzhou Greenchem ILS, LIPC, CAS (Lanzhou, China). Ammonia solution (AR, 25%), hydrochloric acid (AR, 36.5%), chloroform (AR), methanol (AR) and ethanol (AR) were obtained from Beijing Chemical Works. Sodium nitrite (98%) was obtained from Alfa Aesar. Polydimethylsiloxane (PDMS, Sylgard-184) was purchased from Dow Corning Cooperation. Deionized water was obtained by a Milli-Q water-purification system. Self-healing polymer was prepared according to the previous method[25].

**Synthesis of (E)-4-(phenyldiazenyl) phenol**. Sodium nitrite (0.058 mol) was dissolved in deionized water (15 mL) and the solution was dropwise added into a mixture of aniline (0.054 mol) and hydrochloric acid (36.5%, 15 mL) at 0 °C for 5 min to obtain diazonium salt solution. Phenol (0.052 mol) was added to the ammonia-ammonium chloride buffered solution (400 mL, pH = 9) at 0 °C. Then the diazonium solution was dropwise added into phenol solution and stirred vigorously at 0 °C for 3 h. The resulting mixture was mixed with excessive hydrochloric acid (36.5%) and the solid product was collected by filtration. The product (E)-4-(phenyldiazenyl) phenol was then purified by recrystallization in ethanol/water (V/V = 1:1) solution. Finally, the product was dried in a vacuum oven at 60 °C for 24 h. Yield: 47.7%. $^1$H-NMR (400 MHz, CDCl$_3$, δ) 7.855(m, 4 H), 7.491 (m, 3 H), 6.953 (dd, J = 6.95 Hz, 2 H).

**Synthesis of [OMIm]AzoO**. [OMIm]AzoO was prepared by a two-step synthesis. (E)-4-(phenyldiazenyl) phenol (7.838 mmol) and KOH (7.848 mmol) were dissolved in 10 mL ethanol in a nitrogen atmosphere. The mixture was stirred vigorously at 80 °C for 24 h. Then it was filtered and extracted the swirl of filtrate. The product of potassium (E)-4-(phenyldiazenyl) phenolate was dried under vacuum at 60 °C for 24 h. Potassium (E)-4-(phenyldiazenyl) phenolate (0.882 mmol) and [OMIm]Cl (0.882 mmol) were added into ethanol (10 mL) in nitrogen atmosphere. After the mixture was stirred vigorously at 0 °C for 6 h, the product of [OMIm]AzoO was then separated by filtration and purified by recrystallization in ethanol/water (V/V = 1:1) solution. The final product was dried under vacuum at 60 °C for 24 h. Yield: 82.1%. $^1$H-NMR (400 MHz, CDCl$_3$, δ) 7.769(q, 3 H, Ar H), 7.461 (t, J = 7.43 Hz, 2 H), 7.322 (t, J = 7.32 Hz, 1 H), 7.091 (d, J = 7.09 Hz, 2 H), 6.784 (d, J = 6.78 Hz, 2 H), 4.103 (d, J = 4.11 Hz, 2 H), 3.905 (s, 1H), 1.786 (q, 2 H), 1.211 (m, 12 H), 0.848 (t, J = 0.84 Hz, 3 H).

**Preparation of single crystal of [OMIm]AzoO**. The [OMIm]AzoO (40 mg) was dissolved in CH$_2$Cl$_2$ (0.15 mL) and the solution was added into an NMR tube. After C$_6$H$_6$ (0.40 mL) was dropwise added, the NMR tube was sealed and kept at room temperature for 2 weeks, leading to the growth of a needle-like single crystal of [OMIm]AzoO with the size of several millimeters.

**Preparation of crystal-confined ionic liquids**. Typically, the mixture of [OMIm]AzoO (200.0 mg) and [OMIm]PF$_6$ (800.0 mg) was heated to 70 °C. After stirring for 2 min, the binary system was cooled down to room temperature. The final crystal-confined ionic liquids with an addition of 20 wt.% [OMIm]AzoO were obtained. A series of crystal-confined ionic liquids were prepared with the addition of 40 wt.%, 60 wt.% or 80 wt.% [OMIm]AzoO according to the same procedure.

**Characterization**. $^1$H-NMR spectra were recorded on a Bruker spectrometer operating at 400 MHz. X-ray single diffraction was measured through Bruker D8 VENTURE X-ray diffractometer performing ø- and ω-scans (150 (2) K, Mo Kα radiation). Powder X-Ray Diffraction (XRD, SHIMADZU XRD-7000) was used to analyze the crystallization of [OMIm]AzoO. Differential scanning calorimetry (DSC) was used to test the melting point on a DSC822 (METTLER TOLEDO) apparatus. Two heating cycles between 20 °C and 80 °C were measured at a heating speed of 10 °C/min. A thermal-gravimetric analysis was operated by a Q50 TGA (TA Instruments). The self-healing process at micron-size scale was recorded by an optical microscope (Olympus PEN mini E-PMI digital camera) and Atomic Force Microscope (Cyper VRS). Rheological properties of CCILs were characterized using a rheometer (TA ARES-G2). The storage modulus (G') and loss modulus (G'') were assessed by applying a constant strain (5%) over a frequency range from 100 to 0.1 Hz. The reconfiguration was taken out through the oven JING HONG XMTD-8222 at 70 °C for 2 h. The electronic measurements were conducted on a CHI660E electrochemical workstation with a fixed voltage of 1.0 V. Optical images were captured by an Olympus PEN mini E-PMI digital camera. Confocal images were captured by confocal fluorescence microscope (Leica TCS-SP5). Contact angle and surface tension were measured by a drop shape analyzer DSA30, KRÜSS GmbH. The 5 mL syringe with a 23-gauge needle was commercially attained by JIANGSU ZHIYU MEDICAL INSTRUMENT CO., LTD. All basic electronic components were purchased from Beijing Zhongfa Feixun Network Technology Development Co., Ltd and were assembled into complete circuits that possess automatic temperature control.

## Data availability

The data supporting the findings of this work are available within the paper and its Supplementary Information files and from the corresponding authors on request. CCDC 1880207 contains the supplementary crystallographic data for ionic liquid

crystal [OMIm]AzoO. These data can be obtained free of charge from The Cambridge Crystallographic Data Centre.

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

## Acknowledgements

This work was financially supported by the National Natural Science Foundation of China (21825503, 21674127, and 51373197) and the Beijing Municipal National Science Foundation (KZ201610020016). Dr. Fang Liu and Dr. Zhiwen Liu from OXFORD INSTRUMENTS are acknowledged for their help on the AFM test.

## Author contributions

All authors discussed the results and revised on the manuscript. N.G. led the project design and experiments under the supervision of Prof. Y.W.; Y.H. gave helpful advice to fabricate the devices and analyze the data; X.T. helped to deal with the diagrammatic sketch; X.X. and X.W. gave help in the synthesis of ionic liquids.

**Additional information**

**Competing interests:** The authors declare no competing interests.

