## [Peer Review File · Nature Communications]

Reviewers' comments:

Reviewer #1 (Remarks to the Author):

This manuscript entitled with "Crystal-confined Freestanding Ionic Liquids for Reconfigurable and Repairable Electronics" has been submitted to Nature Communications for the consideration of publication. It reports the the usage of a mixture of two kinds of ionic liquids that containing same cations for reconfigurable and repairable electronics. The experiment was carefully designed and the results were analyzed systemically. However, such a paper should not be reconsidered for publication in its current edition because of the following issues.

1. The so-called reconfiguration behavior (Fig. 4) was realized by a heating treatment. It is well known that the melting point of ionic liquid increases with increasing ion size, and many ionic liquids exist in power or wax forms. The reconfigurable behaviors should arise from the melting and condensation characteristics of formulated mixture. It is general that a fluid adapts to any molds only upon gravity force, and droplets tend to coalesce into bigger ones in order to decreasing the total surface energy. Elastic properties of this solid sample should be studied. This reconfiguration behavior is far from the traditional one and it is intrinsic behavior of many ionic liquids with relatively bigger ion size.
2. The authors claim that their CCILs are repairable, as illustrate in Fig.3. However, the self-healing experiment was conducted in a self-healing polymer tube. Generally, the so-called repairable behavior is assigned to the compounds with novel adaptable network based on the reversible reaction (Adv. Mater. 2014, 3561-3566). It will be confusing to call the coalesce of solid-fluid materials to be repairation. Stretch experiment of only CCILs belt or wire should be conducted if the authors want to prove the special self-healing and flexible behavior. In addition, the self-healed sample should show comparable tensile strength.
3. The ionic liquids tend to be confined into nanostructures (Nano Lett. 2015, 15, 3398–3402). Up to 1000% ionic liquid was reported to be confined into BN nanosheets (small 2016, 12, 3535–3542). These nanocomposites are all free-standing and show interesting mechanical performance. All the free standing performances reported here have been studied in molds and tubes. This free standing is misunderstanding.
4. The successful self-healing can also be confirmed by the ion conductivity experiment considering that the sample is a mixture of two kinds of ionic liquids. The author should present more details about resistance study, ionic resistance or electric resistance. The resistance is about 3-8 M Ω , and the resistance change with time. Such behaviors may arise from polarization effect of ion double layers on the electrode plates. The conductivity should be studied by a alternating current. In addition, such a mixture should exhibit a high ion conductivity about 0.1~1 mS/cm. This result cannot support the authors' conclusion.
5. In Fig. 5, the authors show that the robotic arm composed of CCILs can heal itself at high temperature after the arm was break. However, the self-healing mechanism of the CCILs is still unclear, which is important to the rational design and management of the composed materials. Could the author provide more detailed structural information of the CCILs when suffering injury and healing itself at high temperature to discuss the self-healing mechanism of the materials?

Reviewer #2 (Remarks to the Author):

The paper "Crystal-confined Freestanding Ionic Liquids for Reconfigurable and Repairable Electronics" by Gao et al. presents an experimental work on self healing properties of mixture of Ionic liquids and ionic crystal for repairable electronics. The work is interesting and the claim definitely appealing.

From my point of view however the claim is partially supported by the experimental results. In particular, I have two main concerns about this work:

First, the electrical sensing performance is limited to the measurement of conductance/resistance change and absolute current stability during time. However one of the main characteristic of ILs is their response for oscillating and/or sudden change of the applied voltage. This could be tested by imaginary impedance testing like in capacitors. This would help to understand if the configuration proposed modify the charge dynamics.

Second and most importantly, I have to admit to be not fully convinced by the self-healing performances shown in the paper. It is my opinion that further mechanical testing should be performed. This could be done at micron scale using an atomic force microscope showing the actual healing at a smaller scale. Further tests on indentation should be performed and presented.

Beyond these two main issues, authors should also address another question: what is the smallest scale that can be reasonably achieved with their configuration? For the moment only macroscopic tests have been performed but when we talk about repairable electronics one key aspect is the possibility to go down at the micron or even sub-micron scale. At a scale where confinement plays a major role I am not sure (for the moment) that the electrical performances are not deteriorated and self healing still present.

If the authors can reply to my question publication of the manuscript in Nature Communication can be reconsidered.

Reviewer #3 (Remarks to the Author):

This manuscript presents a binary ionic liquid system to be used as reconfigurable and repairable electronics. The highlight and main contribution of this work is this kind of liquid with unique advantages, i.e., freestanding. Sensing and self-healing performance were conducted to characterize its reconfigurable and repairable properties. The experimental design is ingenious. The quantitative analyses in the manuscript are comprehensive, and the discussion is clearly presented. Overall, I would like to recommend the publication of this work on NC, given that the following concerns are addressed in the revision.

1. In Fig. 3, the authors show that the reproducible on-off electrical responses between room temperature and high temperature. However, the results in Fig. Three only show five loops of the melting and crystallization, which is weak to support the opinion that the repeated melting and crystallization have a negligible impact on the sensing performance. After bearing a large number of loops of melting and crystallization, the structure and property of the binary ionic liquid systems maybe have a dramatic change and even lose the corresponding functionality. Could the authors add a longer test of the on-off electrical responses (such as 50 or 100 loops) to explore the potential effect of melting and crystallization on the structure and performance of the materials?
2. A lot of researches have reported that ILs can be confined into nanostructures or Nanosheets, which make ILs are freestanding, also with good mechanical performance. How to interpret their difference? Make a comparison of the mechanical properties between them.
3. My biggest concern is the mechanism of self-healing. If the mechanism is clear, this can be used to guide the design this kind of materials rationally. Is it related to bionics? Try to explain.
4. This study should be completed with a comparison to other simulations and/or to experimental materials, such as supramolecular polyelectrolyte hydrogels, of the studied phenomena.

Reviewer #1 (Remarks to the Author):

This manuscript entitled with “Crystal-confined Freestanding Ionic Liquids for Reconfigurable and Repairable Electronics” has been submitted to Nature Communications for the consideration of publication. It reports the the usage of a mixture of two kinds of ionic liquids that containing same cations for reconfigurable and repairable electronics. The experiment was carefully designed and the results were analyzed systemically. However, such a paper should not be reconsidered for publication in its current edition because of the following issues.

Reply to Reviewer 1:

We are very grateful for your insightful and helpful comments to our manuscript. Our point-to-point responses to all your concerns are as follows:

Q1: The so-called reconfiguration behavior (Fig. 4) was realized by a heating treatment. It is well known that the melting point of ionic liquid increases with increasing ion size, and many ionic liquids exist in powder or wax forms. The reconfigurable behaviors should arise from the melting and condensation characteristics of formulated mixture. It is general that a fluid adapts to any molds only upon gravity force, and droplets tend to coalesce into bigger ones in order to decreasing the total surface energy. Elastic properties of this solid sample should be studied. This reconfiguration behavior is far from the traditional one and it is intrinsic behavior of many ionic liquids with relatively bigger ion size.

[Reply to Q1] Thank you for your critical yet helpful comments on the reconfiguration issue. As you addressed, the size of ions indeed has a significant impact on the condensation characteristics of ionic liquids. However, the crystal-confined phenomenon, which is the key characteristic for the reconfigurable and other important performance in our work, is hardly observed in common binary-phase ionic liquids. In fact, we have similar doubts with you at the beginning of this work. We prepared different binary-phase ionic liquids by mixing [OMIm][PF₆] with other ionic liquids with different ion sizes and ion species (Figure R1a). To be specific, we selected four different solid ionic liquids ([BMIm]Cl, [VEIm]PF₆, [EMIm]PF₆, [VEIm]Br) that are commonly used (Figure R1b-e), and they are all completely nonconductive at room temperature owing to the lack of free moving carriers (Figure R1n). Unfortunately, none of them led to the crystal-confined phenomenon upon mixing with [OMIm][PF₆], and practically, they are either mixed

solution or phase-separation product. As illustrated in Figure R1, [BMIm]Cl, [VEIm]PF₆, and [EMIm]PF₆ are fully dissolved in [OMIm][PF₆] at the loading amount of 60 wt.% or 80 wt.% (Figure R1f-h and Figure R1j-l). These liquid products are conductive while the crystal-like or amorphous solid components that may provide pinning capillary forces are not formed in them. On the contrary, the binary-phase ionic liquids by mixing [OMIm][PF₆] with [VEIm]Br (80 wt.%) are phase separated. It is visually a viscous and non-flowing sample. [VEIm]Br should serve as a form of solid filler to immobilize [OMIm][PF₆]. Such a mixture is non-conductive since too much solid fillers may cut off the ionic pathways (Figure R1i, 1o). However, when the content of [VEIm]Br decreases to 60 wt.%, the gravity-induced stratification becomes dominated because the number of [VEIm]Br fillers is not enough to confine the whole liquid system, and once again, the top liquid layer is conductive as ions are liberated (Figure R1m, 1p). According to these studies, it is not easy to generate a non-flowing binary ionic liquid system that still preserves liquid properties. In principle, a three-dimension network is essentially needed to hinder the fluidic performance of a specific ionic liquid, e.g. [OMIm]PF₆. The fillers for generating three-dimension networks should be reconfigurable, like reversible melting and crystallization as noted, otherwise, the system will not be recycled though it possesses non-flowing property and reasonable conductivity. Moreover, the filler material should have ideal compatibility with [OMIm]PF₆ based on electrostatic interaction, hydrogen bonding interaction or other non-covalent interactions to reduce the serious phase separation. We fully agree with you that [OMIm]PF₆/[OMIm][AzoO] may be not the only formula to prepare a free-standing liquid products. Based on your comments, efforts on exploiting other new building blocks to expand this concept are coming into focus in our laboratory. For your convenience, the corresponding discussions have been added into the revised manuscript in the part of “Discussion”. For your suggestions in regard to the elastic measurements, the binary ionic liquid mixture loaded with 60 wt.% [OMIm][AzoO] was carefully studied by rheology tests. Accordingly, its storage modulus and loss modulus are measured as 1550.99 Pa and 8480.58 Pa, respectively. Since you raised a similar concern about the mechanical

properties in Question 2, more detailed discussions are supplied as below.

Figure R1. Binary-phase ionic liquids consisting of [OMIm]PF₆ and other solid ionic liquids with different mass ratios. **a** Preparation of CCILs as a complex of [OMIm][PF₆] and solid ionic liquids through a super-saturated solution cooling method. Solid ionic liquids at different ratios of binary-phase ionic liquids as follows: **b** [BMIm]Cl (100 wt.%); **c** [VEIm]PF₆ (100 wt.%); **d** [EMIm]PF₆ (100 wt.%); **e** [VEIm]Br (100 wt.%); **f** [BMIm]Cl (80 wt.%); **g** [VEIm] PF₆ (80 wt.%); **h** [EMIm] PF₆ (80 wt.%); **i** [VEIm] PF₆ (80 wt.%); **j** [BMIm]Cl (60 wt.%); **k** [VEIm] PF₆ (60 wt.%); **l** [EMIm] PF₆ (60 wt.%); **m** [VEIm] PF₆ (60 wt.%); **n** Resistance of Sample b~e; **o** Resistance of Sample f~i; **p** Resistance of Sample j~m.

Q2: The authors claim that their CCILs are repairable, as illustrate in Fig.3. However, the self-healing experiment was conducted in a self-healing polymer tube. Generally, the so-called repairable behavior is assigned to the compounds with novel adaptable network based on the reversible reaction (Adv. Mater. 2014, 3561-3566). It will be confusing to call the coalesce of solid-fluid materials to be repairation. Stretch experiment of only CCILs belt or wire should be conducted if the authors want to prove the special self-healing and flexible behavior. In addition, the self-healed sample should show comparable tensile strength.

[Reply to Q2] Thank you for your comments. We have carefully read the paper you provided. To our knowledge, the mechanism of self-healing has been extended from microcapsule strategies to supramolecular methods based on reversible interactions, such as the hetero Diels-Alder reaction in the paper as you noted. However, other non-covalent interactions can also be used as the driving forces to fabricate self-healing materials, such as hydrogen bonding interaction, electrostatic interaction, π - π interaction, and metal-liquid coordination. Besides, growing attentions were also paid to liquid materials for self-healing applications, especially as self-healing electronics. It is one of research interests in our group. Successful examples of self-healing circuits based on liquid metals and self-healing sensors based on ionic liquids or green electrolytes were developed, via encapsulating those conductive liquids into self-healing polymer matrix. In our opinion, self-healing ability of liquid materials is attributed to relatively weak noncovalent interactions, including van der Waals force, hydrogen-bonding and dipole-dipole interactions. These interactions are practically not strong enough to overcome the gravity effect that allow them to be free-standing. This work has made a breakthrough in the area of liquid electronics because the liquid sensing material is able to keep shapes without any encapsulation. According to your comments, more studies on freestanding performance were supplied. Freestanding samples without encapsulation layers and their self-healing behaviors were exemplified in the replies to Question 3 and Figure 5. Tensile stretching of CCILs belt or wire containing 60 wt.% [OMIm][AzoO] was not succeeded because we failed to fix the ionic liquids on the stretcher. However, we believe this failure will not hamper its practical applications as most circuits are

deposited on robust substrates. In order to quantitatively identify the mechanical performance of CCILs, its storage modulus (G') and loss modulus (G'') of CCILs with different $[\text{OMIm}]\text{PF}_6/[\text{OMIm}][\text{AzoO}]$ ratios were evaluated by rheology tests instead of tensile stretching. In addition, rheological behavior after melting and condensation was also characterized to verify the self-healing behavior of CCILs. As shown in Figure R2, the loss modulus of CCILs is always larger than the storage modulus, even for the pure $[\text{OMIm}][\text{AzoO}]$. The rheological result indicates that the viscous flow is superior to elastic deformation and energy will be quickly dissipated once these samples are imposed by mechanical forces. In this regard, CCILs should be assigned to liquid products. For pure $[\text{OMIm}][\text{AzoO}]$, the higher loss modulus than storage modulus is attributed to the slippage of crystal fibers against external forces. It should be noted that rheological behavior kept same after the CCILs samples were melted and recrystallized, fully agreeing with the self-healing and recyclable studies. More discussions about mechanical performance will be continued in the replies to Question 3.

Figure R2. Rheology tests of CCILs at different mass ratio before (Plot: 1) and after (Plot: 2) reconfiguration. **a** 20 wt.%; **b** 40 wt.%; **c** 60 wt.%; **d** 80 wt.%; **e** 100 wt.%.

Q3: The ionic liquids tend to be confined into nanostructures (Nano Lett. 2015, 15,

3398–3402). Up to 1000% ionic liquid was reported to be confined into BN nanosheets (small 2016, 12, 3535–3542). These nanocomposites are all free-standing and show interesting mechanical performance. All the free standing performances reported here have been studied in molds and tubes. This free standing is misunderstanding.

[Reply to Q3] It is really exciting that freestanding and interesting mechanical performances were both achieved by complexing ionic liquids with condensed hollow silica NPs or BN nanosheets. The noted two articles have been cited in the revised manuscript. In order to further convince the freestanding performance, CCILs containing 60 wt.% [OMIm][AzoO] were molded into a sample with human-like shape. As demonstrated in Figure R3, the shape is evidently freestanding without any deformation when it is placed on the bench or grabbed in hand. The freestanding phenomenon means that the improved intermolecular interaction based on crystal confinement is strong enough to support the gravity of ionic liquids. As stated in the manuscript, we have been focusing on developing liquid electronics with the use of liquid sensing materials. The main intention of this manuscript is to solve the leakage problem while still preserving the advantages of liquid materials. It was proved that CCILs should be a liquid-like product according to the rheology studies in Figure R2, which explains the ease of molding and remolding, high flexibility, and self-healable ability. If you allow us to compare our CCILs with the noted two examples, we may say their principles are similar because the confinement is all based on capillary effect. However, the precise formulations are completely different. Ionic liquids confined within HS NPs or BN nanosheets could be considered as examples of heterogeneous complexation. Mechanical property, e.g. represented by freestanding performance in those two papers, was remarkably improved owing to the robust supporting matrix. Yet the heterogeneous complexation is lack of self-healing and reconfiguration properties because those solid matrix is not easy to be reconstructed. In comparison, CCILs are fully composed of ionic liquids. Two components have the same cations which ensures the substantial interfacial interactions. In this case, the binary ionic liquid system stops flowing in the presence of [OMIm][AzoO] less than 5.0 wt.%, though those crystal fibers rarely form cross-linked or condensed networks. If you

refer the good mechanical performance to robust materials with higher storage modulus than loss modulus, our CCILs belong to liquid-like materials with higher loss modulus than storage modulus. The CCILs are expected to possess all the advantages of liquids yet without having leakage problems. In practical applications, instead of sealing in the capillary tubes, CCILs could be also painted on arbitrary substrates to endure mechanical manipulations. We hope these explanations may convince you about the freestanding performance of CCILs and also hope you may give us positive supports on this new area of liquid electronics.

Figure R3. Photos of CCILs (60 wt.%). **a** CCILs molded in a PDMS template with human-like shape; **b** The human-like CCILs is taken from PDMS mold and placed on the bench; **c** The human-like CCILs is grabbed in hand.

Q4: The successful self-healing can also be confirmed by the ion conductivity experiment considering that the sample is a mixture of two kinds of ionic liquids. The author should present more details about resistance study, ionic resistance or electric resistance. The resistance is about 3-8 M Ω , and the resistance change with time. Such behaviors may arise from polarization effect of ion double layers on the electrode plates. The conductivity should be studied by a alternating current. In addition, such a mixture should exhibit a high ion conductivity about 0.1 ~ 1 mS/cm. This result cannot support the authors' conclusion.

[Reply to Q4] Thank you for your helpful concerns on self-healing mechanism. In the practical experiments, the ionic resistance was electrically monitored under an alternating voltage rather than direct voltage. Yes, the resistance gradually increases if the CCILs sample is applied by a direct voltage (Figure R4a). Such a change is attributed to the existence of electrode polarization of ionic double layers as you noted. However, polarization effect can be completely avoided with the use of an alternative mode (As shown in Figure R4b, the resistance of CCILs keeps the same at an alternating voltage (Figure R4b). In this regard, the change of resistance is actually resulted from the temperature change rather than time in the process of sensing tests. Please refer to the corresponding demonstrations in the caption of Figure 3c. In order to address your concern about ionic conductivity, we measured the resistance of CCILs at different ratios through alternating current and made evaluation on the conductivity of CCILs (Figure R4c). The conductivity of pure [OMIm]PF₆ is estimated as 0.56 mS/cm. The conductivity of CCILs decreases upon the addition of [OMIm]AzoO crystals. For CCILs containing 60 wt.% [OMIm]AzoO which is typically used as the sensing material, its conductivity is only 0.05 mS/cm which is at the conductive range of liquid semiconductor.

Figure R4. Electrical test of CCILs on an Electrochemical Workstation. **a** Current measurement of CCILs (60 wt.%) at 25 °C in Amperometric i-t curve mode. **b** Resistance measurement of CCILs (60 wt.%) at 25 °C in Impedance-Time mode. **c** Conductivity curve of CCILs (0 wt.%, 20 wt.%, 40 wt.%, 60 wt.%) at 25 °C in Impedance-Time mode.

Q5: In Fig. 5, the authors show that the robotic arm composed of CCILs can heal

itself at high temperature after the arm was break. However, the self-healing mechanism of the CCILs is still unclear, which is important to the rational design and management of the composed materials. Could the author provide more detailed structural information of the CCILs when suffering injury and healing itself at high temperature to discuss the self-healing mechanism of the materials?

[Reply to Q5] In order to discern the self-healing mechanism of CCILs, a typical self-healing process of CCILs within a specific area was recorded by microscope since suffering an injury till the complete healing at appropriate temperature. As shown in Figure R5a and R5b, at the moment of suffering injury, the ionic pathway was cut off due to the disconnection of [OMIm]AzoO crystals, which initially serve as capillary fillers to confine and bridge [OMIm]PF₆. Notably, the self-healing process began with the migration of ionic liquid of [OMIm]PF₆ into the wounded area (Figure R5c and R5d). This self-motivated flowing phenomenon is attributed to the weakened confinement of [OMIm]AzoO crystal fibers when they are partially melted at an increased temperature. In the following stage, more and more ionic liquids, driven by the melting process of [OMIm]AzoO crystal fibers, were directed to the wounded area (Figure R5e-g). Meanwhile, some unmelted [OMIm]AzoO crystals were also flowed into the wounded area together with [OMIm]PF₆. These migrated crystal fibers acted as “seed” crystals, leading to the growth and regeneration of [OMIm]AzoO crystal fibers in the wounded area. The crystallization of [OMIm]AzoO was accelerated when the temperature was dropped to room temperature (Figure R5h and R5i) and the increased number of [OMIm]AzoO crystal fibers recovered to confine [OMIm]PF₆. At this stage, [OMIm]PF₆ stopped flowing and the broken ionic pathway was fully healed. On the basis of in-situ microscopy observation of the self-healing process of CCILs, it is considered that the self-healing process consists of two main steps. The first step is the migration of [OMIm]PF₆ together with some [OMIm]AzoO crystal fibers from bulk phase to the wounded area. The second step is the recrystallization of [OMIm]AzoO crystals which confine [OMIm]PF₆ and repair the wounded area.

Figure R5. Microscope image of CCILs in a local area in self-healing process. **a** Original sample (25 °C); **b** Damaged sample (25 °C); **c~g** Raising temperature (60 °C) ; **h~i** Cooling down to room temperature (25 °C). Scale bar: 100 μm.

Reviewer #2 (Remarks to the Author):

The paper "Crystal-confined Freestanding Ionic Liquids for Reconfigurable and Repairable Electronics" by Gao et al. presents an experimental work on self healing properties of mixture of Ionic liquids and ionic crystal for repairable electronics. The work is interesting and the claim definitely appealing.

From my point of view however the claim is partially supported by the experimental results. In particular, I have two main concerns about this work:

Reply to Reviewer 2:

We are very grateful for your encouraging comments to our manuscript. Our point-to-point responses to all your concerns are as follows:

Q1: First, the electrical sensing performance is limited to the measurement of conductance/resistance change and absolute current stability during time. However one of the main characteristic of ILs is their response for oscillating and/or sudden change of the applied voltage. This could be tested by imaginary impedance testing like in capacitors. This would help to understand if the configuration proposed modify the charge dynamics.

[Reply to Q1] Thank you very much for your helpful suggestions. As suggested, we measured the resistance of CCILs (60 wt.% [OMIm]AzoO) by imaginary impedance test. According to the testing results, both Nyquist plot and Bode plot of CCILs at different frequency were obtained. As illustrated in Figure R6a, the CCILs possess both impedance and capacitive reactance. According to Figure R6b, the phase of CCILs is gradually close to 90° at high frequency, indicating that the capacitive reactance is the main performance of CCILs at high frequency. In contrast, the impedance reactance becomes the dominated behavior of CCILs when the phase gradually decreases at low frequency. Therefore, it is assumed that our electrical sensors based on CCILs behave more like resistors rather than capacitors. Besides, the imaginary impedance tests of reconfigured CCILs after heating and cooling treatment were also accomplished. Notably, the resistance of reconfigured CCILs is almost the same with original resistance. Thank you for your advice once again, and it is really helpful for understanding the reconfiguration behavior. Corresponding discussions have been added to the manuscript. Moreover, electrical performance of CCILs at different frequency was also supplied by using the method of IMPT mode.

Figure R6. Electrical test of CCILs (60 wt.% [OMIm]AzoO) at different frequency. **a** Nyquist plot of CCILs (60 wt.% [OMIm]AzoO) (Black plot: original sensing chip;

Red plot: reconfigured sensing chip). **b** Bode plot of CCILs (60 wt.% [OMIm]AzoO) (Black plot: original sensing chip; Red plot: reconfigured sensing chip).

Q2: Second and most importantly, i have to admit to be not fully convinced by the self-healing performances shown in the paper. It is my opinion that further mechanical testing should be performed. This could be done at micron scale using an atomic force microscope showing the actual healing at a smaller scale. Further tests on indentation should be performed and presented.

[Reply to Q2] As suggested, mechanical tests and the study of self-healing behavior at the micro scale were attempted. As shown in Figure R7, the mechanical performance of CCILs was quantitatively identified according to storage modulus (G') and loss modulus (G'') by rheology tests. In addition, rheological behavior after melting and condensation was also characterized to verify the self-healing behavior of CCILs. The loss modulus of CCILs is always larger than the storage modulus, even for the pure [OMIm][AzoO]. The rheological result indicates that the viscous flow is superior to elastic deformation and energy will be quickly dissipated once these samples are imposed by mechanical forces. In this regard, CCILs are confirmed as a particular class of liquid products. For pure [OMIm][AzoO], the higher loss modulus than storage modulus is attributed to the slippage of crystal fibers against external forces. It should be noted that rheological behavior keeps same after the CCILs samples were melted and reconfigured, fully agreeing with the self-healing and recyclable studies.

The studies of self-healing process by atomic force microscope at a micron scale were supplied in Figure R8. Typically, a mechanical damage with a width about 1 μm on the surface of CCILs (60 wt.% [OMIm][AzoO]) was scratched by the AFM tip. Similar to the microscopy observation of self-healing process at larger scale (Figure R9, more detailed discussion please refer to the replies to Q5 for referee 1), a little [OMIm]PF₆ component quickly flows into the damaged groove once the mechanical damage is appeared. The faster liquid migration at smaller scale is due to the enhanced capillary effect by the damaged groove which is like an opened capillary

tube. However, solely [OMIm]PF₆ can partially fill the damaged groove because [OMIm]PF₆ is also confined by [OMIm][AzoO] crystal fibers nearby. Upon increasing the temperature from 5 °C to 50 °C, CCILs as a mixture of [OMIm]PF₆ and [OMIm][AzoO] would be gradually loosen as a result of melting of [OMIm][AzoO] crystal fibers (Figure R8c-k). More ionic liquids including both [OMIm]PF₆ and [OMIm][AzoO] are driven into the damaged part, fully filling the damaged groove. Once the temperature drops to room temperature, CCILs were recondensed and the damaged part was healed (Figure R8l). Overall, the self-healing process at a scale of 1 μm (Figure R8b) is similar to the healing process at a larger scale of 200 μm (Figure R9b). They both contain two steps, including the migration process of CCILs and recrystallization of [OMIm][AzoO] crystal fibers. According to the time scale, mechanical damages with smaller size can be healed more rapidly owing to the enhanced capillary effect. Thank you for your helpful suggestions once again. The studies on self-healing at a small scale are definitely beneficial for understanding the self-healing mechanism. Corresponding discussions were added and marked with yellow background in the revised manuscript.

Figure R7. Rheology tests of CCILs at different mass ratio before (Plot: 1) and after (Plot: 2) reconfiguration. **a** 20 wt.%; **b** 40 wt.%; **c** 60 wt.%; **d** 80 wt.%; **e** 100 wt.%. **f** Viscosity test of CCILs with different content of [OMIm]AzoO.

Figure R8. AFM observations of self-healing process of CCILs (60 wt.% [OMIm]AzoO). **a** Original sample (5 °C); **b** Destroyed surface of sample with AFM tip (5 °C); **c~k** Raising temperature to 10 °C, 15 °C, 20 °C, 25 °C, 30 °C, 35 °C, 40 °C, 45 °C, 50 °C gradually; **l** Cooling down to room temperature (25 °C). Scale bar: 3 μm .

Figure R9. Microscope image of CCILs in a local area in self-healing process. **a** Original sample (25 °C); **b** Damaged sample (25 °C); **c~g** Raising temperature (60

°C); **h-i** Cooling down to room temperature (25 °C). Scale bar: 100 μm.

Q3: Beyond these two main issues, authors should also address another question: what is the smallest scale that can be reasonably achieved with their configuration? For the moment only macroscopic tests have been performed but when we talk about repairable electronics one key aspect is the possibility to go down at the micron or even sub-micron scale. At a scale where confinement plays a major role I am not sure (for the moment) that the electrical performances are not deteriorated and self healing still present.

[Reply to Q3] Thank you for your helpful and important suggestion. We apologize that we previously paid little attention to this important concern. This suggestion is believed very helpful for us to expand our work from macroscopic electronics to microscopic electronics. As stated in the response to your second question, AFM study as well as microscopy study convinces the success of self-healing process at micron scales. In order to identify if electrical performance is preserved at those smaller scales, CCILs were put into a series of rectangular channels with the same depth of 50 μm but different width of 20 μm, 30 μm, 40 μm, 50 μm, 100 μm, or 200 μm to serve as micron-size electrical chips. Electrical tests revealed that the chips with width of 20 μm, 30 μm, 40 μm, 50 μm, and 100 μm are not appropriate for method of IMPT mode, because their resistance are all higher than the measurement limitation of 100 MΩ by the electrochemical workstation. Exceptionally, the chip with width of 200 μm has a reasonable resistance of 79.43 MΩ, which fits IMPT requirements and also meets the demands for sensing application. According to electrical tests in Figure R10a, mechanical bending has negligible influence on the resistance of this chip as the resistance keeps almost the same at arbitrary bending angle. Additionally, the electrical self-healing behavior of this chip was also evaluated. As illustrated in Figure R10b, the resistance can completely recover to its initial level after repeated breaking and healing processes. In addition, the temperature sensing ability can be also recovered after the self-healing process (Figure R10c). The relative resistance change against the temperature change from 25 °C to 40 °C is almost the same after the sensor is broken and repaired for three times. Further microscopy

observations of the self-healing process in the micro-channel are fully consistent with the AFM observation. Accordingly, the ionic liquid of [OMIm]PF₆ flows into the damaged area upon increasing the temperature (Figure R10d-i, ii). Afterwards, [OMIm]AzoO crystals were carried into the same position by [OMIm]PF₆ flowing (Figure R10d-iii), followed with crystal melting under higher temperature as well as thorough mixing of [OMIm]AzoO with [OMIm]PF₆ (Figure R10d-iv). After the recrystallization of [OMIm]AzoO, CCILs were reformed and the damaged area was fully healed at room temperature (Figure R10d-v~viii). Based on the electrical studies and self-healing investigation at micron scale, it is believed the concept of freestanding sensing liquid could be also extended to microelectronics.

Figure R10. Self-healing measurement of CCILs (60 wt.% [OMIm]AzoO) at micron scale (size: 200 μm), setting the measurement length as 1 cm. **a** Relative resistance at different bending angles. **b** Cycles of electrical repairing after the electrical chip is damaged and repaired for multiple times. (Orange area: Off-state circuits after being cut. Blue area: On-state circuits after repairing.) **c** Cycles of resistance change against temperature change from 25 $^{\circ}\text{C}$ to 40 $^{\circ}\text{C}$. **d** Microscopy images of self-healing process

of CCILs (60 wt.% [OMIm]AzoO) in a rectangular channel with width of 200 μm .
Scale bar: 100 μm .

Reviewer #3 (Remarks to the Author):

This manuscript presents a binary ionic liquid system to be used as reconfigurable and repairable electronics. The highlight and main contribution of this work is this kind of liquid with unique advantages, i.e., freestanding. Sensing and self-healing performance were conducted to characterize its reconfigurable and repairable properties. The experimental design is ingenious. The quantitative analyses in the manuscript are comprehensive, and the discussion is clearly presented. Overall, I would like to recommend the publication of this work on NC, given that the following concerns are addressed in the revision.

Reply to Reviewer 3:

We are very grateful for your helpful and positive comments to our manuscript. Our point-to-point responses to all your concerns are as follows:

Q1: In Fig. 3, the authors show that the reproducible on-off electrical responses between room temperature and high temperature. However, the results in Fig. Three only show five loops of the melting and crystallization, which is weak to support the opinion that the repeated melting and crystallization have a negligible impact on the sensing performance. After bearing a large number of loops of melting and crystallization, the structure and property of the binary ionic liquid systems maybe have a dramatic change and even lose the corresponding functionality. Could the authors add a longer test of the on-off electrical responses (such as 50 or 100 loops) to explore the potential effect of melting and crystallization on the structure and performance of the materials?

[Reply to Q1] Thank you for your helpful suggestion. As suggested, 100 cycles of the temperature sensing tests of CCILs with different content of [OMIm]AzoO are summarized in Figure R11. Remarkably, CCILs exhibit reliable and repeatable sensitivity against temperature change although they have been engaged with a great number of melting and recrystallization. This study is very important for us to confirm the negligible influence of melting and crystallization on the electrical responses of

CCILs and also offers great promise for long-term service. The supplied new data and corresponding discussions have been added in the revised manuscript.

Figure R11. 100 cycles of temperature sensing test of CCILs with different content of [OMIm]AzoO. **a** CCILs (0 wt.% [OMIm]AzoO); **b** CCILs (20 wt.% [OMIm]AzoO); **c** CCILs (40 wt.% [OMIm]AzoO); **d** CCILs (60 wt.% [OMIm]AzoO).

Q2: A lot of researches have reported that ILs can be confined into nanostructures or Nanosheets, which make ILs are freestanding, also with good mechanical performance. How to interpret their difference? Make a comparison of the mechanical properties between them.

[Reply to Q2] We fully agree with your opinion about the confinement of ionic liquids by other nanostructures or nanosheets for formulating ionic liquids as freestanding materials with reasonable mechanical performance. For your

convenience, some successful examples and the typical CCILs with 60 wt.% [OMIm]AzoO are summarized and compared in Table R1. Four aspects are especially compared, including elastic modulus, freestanding performance, self-healing performance, and reconfigurable performance. The former two aspects are mainly based on the solid supporting matrix, while the latter two mainly rely on the intrinsic properties of ionic liquids. Covalently cross-linked nanostructures are generally preferred as they are considered as supporting matrix with better mechanical performance, such as networked epoxy resins, PVDF-co-HFP-5545, PMMA-Silica nanocomposites, PMMA skeleton and PVA-ZIF-8 composite membrane. Though some of them were not mentioned if they are freestanding after loading ionic liquids, they are believed robust enough against mechanical manipulation according to their elastic modulus values. Some other condensed nanostructures without covalent crosslinking, e.g. hollow silica spheres and graphene-analogues boron nitride nanosheets, could also provide considerable capillary force to generate freestanding ionic liquid products. These composites, however, hardly possess self-healing and reconfigurable properties because they are difficult to be reorganized and reconstructed once suffering damages. In contrast, composites possessing relatively low elastic modulus are possible to be exploited as self-healing materials, e.g. physically mixing ionic liquids with carbon nanotube, because they are relatively easier to be reorganized. In practice, it is challenging to endow ionic liquids with freestanding, self-healing, and reconfigurable abilities at the same time. This work adopts a strategy of mixing one ionic liquid with another ionic liquid. Two ionic liquids have the same cations which ensures the substantial interfacial interactions. In this case, the binary ionic liquid system stops flowing in the presence of [OMIm][AzoO] less than 5.0 wt.%, though those crystal fibers rarely form cross-linked or condensed networks. The CCILs products are considered as liquid-like materials in terms of higher loss modulus than storage modulus. Upon overcoming the gravity-induced liquid flowing by improving interfacial interactions, this particular class of ionic liquid products are expected to have all the advantages of liquids yet without having leakage problems.

Making the long explanation short, the binary ionic liquid system preserves the nature of liquids while solving the leakage problem. It is therefore able to meet the demands on high flexibility, self-healing and recyclable abilities.

Table R1. Comparison of different composites by confining ionic liquids within different nanostructures.

Property Nanostructure	Elastic Modulus (MPa)	Freestanding Performance	Self-healing Performance	Reconfigurable Performance
Networked Epoxy Resins ¹	45	—	—	—
PVDF-co-HFP-5545 ²	0.1	—	yes	—
PMMA-Silica Nanocompo-sites ³	0.3	—	—	—
Carbon Nanotube ⁴	5.0×10^{-5}	—	yes	—
PMMA Skeleton ⁵	13	—	—	—
PVA-ZIF-8 Composite membrane ⁶	30	—	—	—
Hollow silica spheres ⁷	—	yes	—	—
BN Nanosheets ⁸	—	yes	—	—
[OMIm]AzoO	1.6×10^{-3}	yes	yes	yes

Note: “—” refers to the corresponding property not mentioned.

Reference:

- [1] Matsumoto, K. and Endo T. *Macromolecules* **2008**, *41*, 6981.
- [2] Cao, Y., Morrissey, T. G., Acome, E., Allec, S. I., Wong, B. M., Keplinger, C. and Wang, C. *Adv. Mater.* **2017**, *29*, 1605099.
- [3] Gayet, F., Viau, L., Leroux, F., Mabilie, F., Monge, S., Robin, J-J. and Vioux, A. *Chem. Mater.* **2009**, *21*, 5575.
- [4] Fukushima, T., Kosaka, A., Ishimura, Y., Yamamoto, T., Takigawa, T., Ishii, N. and Aida, T. *Science* **2003**, *300*, 2072.
- [5] Susan, M. A-B-H., Kaneko, T., Noda, A. and Watanabe, M. *J. AM. CHEM. SOC.* **2005**, *127*, 4976.
- [6] Liu, C., Zhang, G., Zhao, C., Li, X., Li M. and Na H. *Chem. Commun.* **2014**, *50*, 14121.
- [7] Zhang, J., Bai, Y., Sun, X.-G., Li, Y., Guo, B., Chen, J., Veith, G. M., Hensley, D. K., Paranthaman, M. P., Goodenough, J. B. and Dai, S. *Nano Lett.* **2015**, *15*, 3398.
- [8] Li, M., Zhu, W., Zhang, P., Chao, Y., He, Q., Yang, B., Li, H., Borisevich, A.

and Dai, S. *Small* **2016**, *12*, 3535.

Q3: My biggest concern is the mechanism of self-healing. If the mechanism is clear, this can be used to guide the design this kind of materials rationally. Is it related to bionics? Try to explain.

[Reply to Q3] The above two referees raised similar concerns about the mechanism of self-healing process. As stated above, microscopy observations at small scale by microscope and AFM illustrate that the self-healing process includes the steps of liquid migration and recrystallization of [OMIm]AzoO. The demonstration of self-healing mechanism is supplied once again for your convenience. According to microscope observation in Figure R12a-b, at the moment of suffering injury, the ionic pathway was cut off due to the disconnection of [OMIm]AzoO crystals, which serve as capillary fillers to confine and bridge [OMIm]PF₆. Notably, the self-healing process began with the migration of ionic liquid of [OMIm]PF₆ into the wounded area (Figure R12c-d). This self-motivated flowing can be attributed to the weakened confinement of [OMIm]AzoO crystal fibers when they are partially melted at an increased temperature. In the following stage, more and more ionic liquids, driven by the melting process of [OMIm]AzoO crystal fibers, were directed to the wounded area (Figure R12e-g). Meanwhile, some unmelted [OMIm]AzoO crystals were also flowed into the wounded area together with [OMIm]PF₆. These migrated crystal fibers acted as “seed” crystals, leading to the growth and regeneration of [OMIm]AzoO crystal fibers in the wounded area. The crystallization of [OMIm]AzoO was accelerated when the temperature was dropped to room temperature (Figure R12h-i) and the increased number of [OMIm]AzoO crystal fibers recovered to confine [OMIm]PF₆. At this stage, [OMIm]PF₆ stopped flowing and the broken ionic pathway was fully healed. Based on the in-situ microscopy observation of the self-healing process of CCILs, it is considered that the self-healing process consists of two main steps. The first step is the migration of [OMIm]PF₆ together with some [OMIm]AzoO crystal fibers from bulk phase to the wounded area. The second step is the recrystallization of [OMIm]AzoO crystals which confine [OMIm]PF₆ and repair the wounded area.

AFM study also gives a similar demonstration on the self-healing mechanism,

though it indicates the self-healing process is faster at the smaller scale. Please refer to the AFM study in the answer to Question 2 for referee 2.

Figure R12. Microscope image of CCILs in a local area in self-healing process. **a** Original sample (25°C); **b** Damaged sample (25°C); **c~g** Raising temperature (60 °C) ; **h~i** Cooling down to room temperature (25°C). Scale bar: 100 μm.

Q4: This study should be completed with a comparison to other simulations and/or to experimental materials, such as supramolecular polyelectrolyte hydrogels, of the studied phenomena.

[Reply to Q4] Thank you for your helpful suggestion. As suggested, many other experimental materials were searched and several formulations are exemplified as below. In addition to nanostructures and nanosheets as mentioned in the answer to Question 2 for referee 1, supramolecular polyelectrolyte hydrogels could also be potentially used as supporting matrix to confine water-based liquid materials. For example, polyacrylamide hydrogel containing sodium chloride was served as ionic conductors for stretchable and transparent strain sensors^[1]. Another hydrogel-based

triboelectric nanogenerator for energy harvesting and self-powered sensors was prepared by using physical-crosslinking polyvinyl alcohol hydrogel as substrate materials^[2]. Recently, a novel supramolecular polyelectrolyte hydrogel, based on PAA-co-DMAPS copolymer swollen in sodium chloride solvent, was demonstrated to adopt a sandwich configuration to detect mechanical deformation and temperature changes, similar to natural skins^[3]. In addition to these examples, there are also many other excellent researches about hydrogels that offer great promises in the field of electronic sensors. However, it is worth noting that hydrogel systems are routinely encountered with the problems of water volatilization at open states, water swelling in highly humidity conditions, or frozen at low temperature. These disadvantages definitely hinder the long-term service of the hydrogel systems in the complex environment.

Ionic gel is a better choice to avoid the problem of solvent volatilization. In the past decade, intensive studies were devoted to ionic gel systems by confining ionic liquids within polymer networks. Thermo-reversible ionic gel with tunable modulus possesses a storage modulus of 10 kPa and considerable conductivity based on the self-assembly of an ABA triblock copolymer within a room-temperature ionic liquid^[4]. Another thermo-reversible ionic gel through the gelation of PNIPAm–PEO–PNIPAm in [EMIM][TFSI] exhibited highly conductive as a viscous fluid^[5]. However, to our knowledge, the study of ionic gels is still at a young stage. Rarely ionic gel systems were extended to sensing applications. In our opinions, they may be compatible for reconfigurable electronics if the reassembly of polymer gelators has little influence on the conductive and sensing performance, meanwhile, keeping the content of gelator at a reasonable value. In these regards, it will be an intriguing and attractive research direction by expanding ionic gels in the electrically sensing applications. In comparison with polymer system, this work merely combines two ionic liquids. It is a system only composed of small molecules. The strong interfacial interaction, combined with rapid recrystallization, ensures the ease of self-healing and recycling operations. We believe that we have made a great step on the use of ionic liquids as electrically sensing materials. The binary ionic liquid system is fluidic yet

freestanding. This seems a contradictory system while it is well explained. In practice, instead of sealing in the capillary tubes, the material be also painted on arbitrary substrates, serving as a special electrical coating.

Once again, we highly appreciate your helpful suggestion. The successful examples hydrogel systems give us valuable inspirations to design more complicated liquid electronics, e.g. exploiting liquid capacitor, liquid inductance, liquid slide rheostat, and liquid diode. According to your suggestions, great efforts will be also paid to ionic gels with the use of stimuli-responsive gelators, aiming to trigger the reconfiguration by other ways.

[1] Sun , J.-Y., Keplinger, C., Whitesides, G. M. and Suo, Z. *Adv. Mater.* **2014**, *26*, 7608.

[2] Xu, W., Huang, L.-B., Wong, M.-C., Chen, L., Bai, G. and Hao, J. *Adv. Energy Mater.* **2016**, *7*, 1601529.

[3] Lei, Z. and Wu, P. *Nat. Commun.* **2018**, *9*, 1134.

[4] Zhang, Y.-D., Fan, X.-H., Shen, Z. and Zhou, Q.-F. *Macromolecules* **2015**, *48*, 4927.

[5] He, Y. and Lodge, T. P. *Chem. Commun.* **2007**, *0*, 2732.

REVIEWERS' COMMENTS:

Reviewer #1 (Remarks to the Author):

The authors have tried their best to answer all the comments and this current form has been improved greatly. However, our most concern on the relationship between self-healing/ self-repairing and the available experiment results is still awaiting for full satisfaction. As commented previously, the so-called self-healing behavior looks more like a melting and condensation process of a mixtures. The authors assigned the self-healing to the weak interactions (van der Waals force, hydrogen-bonding and dipole-dipole interactions). Note that these weak interactions are general in pure ionic liquids or formulated mixtures. The cutting and healing experiment of arm (Fig 6) conducted at varied temperature can not be considered as a healing process. This experiment will be more convincing if the wound can be repaired if this experiment was allowed to happen at room temperature. The absence of direct tensile stretching experiment on pure CCILs belt or wire just implies that this behavior can be assigned mostly to a remolding process.

Reviewer #2 (Remarks to the Author):

The authors have addressed all the questions raised and they have performed additional experiments when required. I am glad to recommend publication of the manuscript in Nature Communications

Reviewer #3 (Remarks to the Author):

This work is interesting with good analysis and well designed experiment. After careful review of the revised manuscript and point to point response, I recommend it to be published by NC.